# An Investigation on a Quantitative Tomographic SHM Technique for a Containment Liner Plate in a Nuclear Power Plant with Guided Wave Mode Selection

**DOI:** 10.3390/s19122819

**Published:** 2019-06-24

**Authors:** Yonghee Lee, Younho Cho

**Affiliations:** 1Graduate school of mechanical system design, Pusan National University, Busan 46241, Korea; leedragon83@pusan.ac.kr; 2School of mechanical engineering, Pusan National University, Busan 46241, Korea

**Keywords:** RAPID (reconstruction algorithm for probabilistic inspection of damage), guided wave, containment linear plate, tomography, finite element simulation, wave mode analysis, SHM

## Abstract

The containment liner plate (CLP) in a nuclear power plant is the most critical part of the structure of a power plant, as it prevents the radioactive contamination of the surrounding area. This paper presents feasibility of structural health monitoring (SHM) and an elastic wave tomography method based on ultrasonic guided waves (GW), for evaluating the integrity of CLP. It aims to check the integrity for a dynamic response to a damaged isotropic structure. The proposed SHM technique relies on sensors and, therefore, it can be placed on the structure permanently and can monitor either passively or actively. For applying this method, a suitable guided wave mode tuning is required to verify wave propagation. A finite element analysis (FEA) is performed to figure out the suitable GW mode for a CLP by considering geometric and material condition. Furthermore, elastic wave tomography technique is modified to evaluate the CLP condition and its visualization. A modified reconstruction algorithm for the probabilistic inspection of damage tomography algorithm is used to quantify corrosion defects in the CLP. The location and shape of the wall-thinning defects are successfully obtained by using elastic GW based SHM. Making full use of verified GW mode to Omni-directional transducer, it can be expected to improve utilization of the SHM based evaluation technique for CLP.

## 1. Introduction

In case of leakage radiation accidents, the protection facilities play an important role in preventing radiation diffusion. The containment liner plate (CLP) is one of the most crucial facilities of a nuclear power plant that requires continuous monitoring and repair. The CLP consists of a cylindrical steel shell that forms the inner wall, a ceiling, and the floor of the reactor containment building, each having a plate shape. It is "one of the many" defense-in-depth barriers that ensures the safety of nuclear energy plants. The CLP is the key protective measure against a nuclear accident and forms the base-structure of the concrete reactor building. To ensure safe operation, CLP structures have to be inspected or monitored, either periodically or in real time, for a potential failure. 

For this purpose, ultrasonic nondestructive evaluation (NDE) techniques have been used extensively. In the nuclear power plant industry, the conventional NDE technique, i.e., the ultrasonic test (UT), is widely applied to detect a failure in CLP. A typical, conventional UT inspects the region of the structure immediately below or adjacent to the transducer(s). This means that, if a large structure is to be tested, the whole surface of the transducer(s) must be scanned, the results for which are commonly termed as B or C-scan. Nowadays, the UT for thickness evaluation has to be conducted for the radioactive surroundings of the facility by an ultrasonic NDE engineer. Consequently, an improved and safer evaluation procedure is required to overcome the high-risk conditions of the working environment. To come up with enhanced solutions ensuring better safety and no human involvement, a possible alternative is the real-time-based SHM. The ultrasonic GW. In the case of using an omnidirectional transducer, the wave power density is not sufficient compared with a unidirectional transmitter way. 

To make up this problem, the proper wave mode selection is also significantly discussed. This GW technique has been used in various industrial fields as an alternative NDE method, especially, for pipe and plate-like structures [1]. The basic concept of the GW inspection technique depends on energy level variation, mode conversion, and attenuation of wave signals throughout the defect region [2]. A theoretical background of a guided ultrasonic wave inspection technique, including Rayleigh surface wave and Lamb wave, has also been studied [3]. The previous research shows us that selecting a suitable wave mode is needed to realize the well-developed SHM algorithm. The characteristic of geometric and material properties are significant factors for GW mode selection, which can generate the dominant GW signal and reconstruct images of defects’ condition on CLP. Generally, the dispersion curves of the Lamb waves are an essential guide for optimizing the mode and frequency [4]. Numerical analyses are also considered as a method to generate a dominant guided wave. Using the boundary element method, mode conversion and suitable GW mode selection are developed for a plate-like structure [5]. From the viewpoint of thickness variation and wall-thinning states, the FEA is the alternative way to verify damages in a plate-like structure [6]. 

The SHM algorithm is also an alternative technique that can overcome the limitations of conventional UT including visualization of the assessment. The ultrasonic tomography detects defects in specimens much more efficiently than signal analysis. The tomography’s theoretical progress has been researched for medical and structural diagnosis based on the X-ray phenomenon application [7]. For developing UT-based inspection, UT-based tomography has also been researched. Combining computing tomography (CT) algorithms with GW features, researchers have been able to generate tomograms that reveal structural integrity of the objects under inspection. The filtered back projection (FBP) algorithm is commonly applied to the visualization method based on weighting, filtering, and back-projection [8]. 

The tomographic algorithm used here is based on a reconstruction algorithm for probabilistic inspection of damage (RAPID). Lamb wave RAPID tomography has been used successfully in a number of applications ranging from material loss verification to penetrated-hole detection [9]. The work to date has been primarily performed using contact transducers [10,11,12,13], air-coupled transducers [14,15], and laser ultrasound [16,17]. Significant literature has been published on Lamb wave pipe inspection applications [18,19,20]. Especially, for the elbow-shape region in a pipe, including bended case, the defects are dramatically inspected on the pipe considering environmental condition [19]. Earlier research showed that qualitative defect imaging with ghost imaging was able to describe the accurate size of the failure. Guided wave tomographic visualization shows great potential in identifying the defect’s location and quantitative sizing [21,22,23,24]. 

The availability of defect tomographic comparison work also has been undertaken. Both FBP and RAPID algorithms result in similar image reconstructions [9]. Defects having simple shapes, such as a circle, are quantitatively illustrated using a variable shape factor [22]. For defects having irregular shapes or in the case of multiple defects, clarifying reconstruction image techniques are studied based on the RAPID algorithm using variable shape factor approach [23]. For the development of highly accurate defects, especially wall-thinning depth, the full wave inversion (FWI) algorithm is significantly studied for corrosion damages and wall-thinning defects [25]. For the alternative technique for enhancing the accuracy of wall-thinning defects, scattering phenomena are also treated. In particular, the HARBUT (The Hybrid Algorithm for Robust Breast Ultrasound Tomography) based algorithm is handled and improved the modified HARBUT technique to verify the defect depth with highly accurate results [26]. The other significant research presents the technique of detection on composite laminate which acts differently from metallic material [27]. In this research, high-risk facilities are treated such as aerospace, nuclear power plants and ocean plant construction. In the case of nuclear power plant facilities, the inspection technique not only relies on the conventional UT but also GW based techniques with reconstructed image algorithms frequently [24].

To come up with an enhancement of the SHM technique, GW based SHM, applications of Omni-directional transducer, is also studied. The ring-type of EMAT practical research is already performed based on Lamb wave propagation and its tomography [28]. The same manner of using PZT, a baseline-free SHM technique, is developed for evaluating complex anisotropic composite media [29]. In addition, the very promising SHM technique, based on generation of SH wave, is designed, which consists of two thickness-poled piezoelectric half-ring connectivity [30].

Note that wall-thinning damage on the plate will cause durability weakness for external impact or disturbance. The quantitative and reliable evaluation technique, GW based SHM, is required to evaluate reliability of facilities. However, the best and most efficient mode choice is unknown due to the lack of knowledge of the actual GW propagation condition, which is usually required in engineering practice. The present study focuses on the verifying suitable GW mode using FEA, wave structure analysis, and tomography applications. First, a simple analytical model is proposed to describe the relation between the characteristics of GW and the wall-thinning damage states of carbon steel plates using wave structure analysis. Next, from a practical viewpoint, GW waves, such as Lamb waves, are generated to verify the appropriate mode using FEA. Finally, the most suitable GW mode is established by comparing the results of various GW modes with tomographic results.

## 2. Theoretical Fundamentals

### 2.1. Guided Wave Analysis for Suitable Mode Selection

The GW study has already been developed and formulated by earlier researchers. In particular, the Lamb wave and Rayleigh surface wave phenomena have been studied by Rose et al., Cawley et al., and Viktorov et al. [1,2,3]. The classical problem of the Lamb wave propagation is associated with the wave motion in a traction-free homogeneous isotropic plate. In the case of Lamb wave modes, two systems are split in symmetric Equation (1) and antisymmetric Equation (2) mode depending on fluctuating patterns, where the variables of ‘*p*’ and ‘*q*’ are as shown in the Equation (3) and ‘*k*’ is the wavenumber: (1)$$\frac{\tan(qh)}{\tan(ph)}=-\frac{4k^{2}pq}{{\left(q^{2}-k^{2}\right)}^{2}},\;\mathrm{for}\;\mathrm{symmetric}\;\mathrm{modes},$$
(2)$$\frac{\tan(qh)}{\tan(ph)}=-\frac{{\left(q^{2}-k^{2}\right)}^{2}}{4k^{2}pq},\;\mathrm{for}\;\mathrm{antisymmetric}\;\mathrm{modes},$$
(3)$$p^{2}=\frac{\omega^{2}}{c_{L}{}^{2}}-k^{2}\;\mathrm{and}\;q^{2}=\frac{\omega^{2}}{c_{T}{}^{2}}-k^{2}.$$

From the dispersion equations, the dispersion curve is as shown in Figure 1. The group velocity is obtained by calculating Equation (4) and its thickness ‘d’ is 6 mm, where , $c_{p}$, $c_{g}$, f, d, ω, and k are Rayleigh surface wave velocity, phase velocity, group velocity, frequency, thickness of plate, angular frequency and wave number, respectively. In this study, a proper mode selection plays an important role in the realization of SHM system based on GW. To generate and select a proper GW mode in the experiment, the guided wave dispersion curve diagram is required. For drawing the dispersion curve, longitudinal and transverse wave speeds (C_L_ = 5850 m/s and C_T_ = 3158 m/s) are measured by experimental work. For the purpose of figuring out generable mode selection, preferentially the experiment about Lamb wave generation is performed. The CLP is relatively much thicker than any other plate treated in many previous research works [6,7,8,9,10,11,12,13,14,15,16,17,19,20,21]. Two types of frequencies are treated to generate lamb waves. Both transducers are based on PZT material that can generate the most powerful way to excitation. The fundamental frequencies of transducers are 500 kHz (available 400 to 600 kHz) and 225 kHz (available 200 to 300 kHz), respectively. The frequency band refers to the bandwidth data from the initial state of the transducer. Higher frequency modes are not suitable due to being easily attenuated with respect to the propagation distance.

To figure out the proper wave modes, there are six GW modes: A0 1.0 MHz·mm, S0 2.0 MHz·mm, A1 3.0 MHz·mm, S1 3.5 MHz·mm, S0 3.0 MHz·mm and A1 3.6 MHz·mm are considered to generate a guided wave mode. Each mode of phase velocity is 2322 m/s (A0 1.0 MHz·mm), 4810 m/s (S0 2.0 MHz·mm), 6010 m/s (A1 3.0 MHz·mm), 5970 m/s (S1 3.5 MHz·mm), 3520 m/s (S0 3.0 MHz·mm) and 5510 m/s (A1 3.6 MHz·mm), respectively. Furthermore, each mode of the group velocity is calculated as 3142 m/s (A0 1.0 MHz·mm), 3266 m/s (S0 2.0 MHz·mm), 4520 m/s (A1 3.0 MHz·mm), 3310 m/s (S1 3.5 MHz·mm), 2080 m/s (S0 3.0 MHz·mm) and 3320 m/s (A1 3.6 MHz·mm), respectively. For ultrasonic tensiometry of a thin plate, symmetric modes that are higher than fundamental order are used, with a heterogeneously stressed plate owing to the dominance of longitudinal particle vibrations [2]:(4)$$c_{g}=\frac{d\omega }{dk}\;\mathrm{and}\;c_{g}=c_{p}{}^{2}{\left[c_{p}-(fd)\frac{dc_{p}}{d\left(fd\right)}\right]}^{-1}.$$

In this paper, the specimen is much thicker than the ones used in other Lamb wave inspection [6,7,8,9,10,11,12,13,14,15,16,17,19,20,21]. It means that the Lamb wave with dominant longitudinal displacement, where displacement or power flux is concentrated on the outside surface, can be the sustainable mode for energy loss and attenuation. From the viewpoint of Lamb wave propagation and the dominant mode selection, symmetric wave, dominance of longitudinal wave, is the appropriate mode for CLP plates. Furthermore, antisymmetric mode, influenced by transverse wave and shear stress, is a bending mode that cannot ensure long-range propagation like the symmetric mode [2]. Lamb wave is a dispersive mode, which means that the wave mode is affected by frequency and geometric conditions. 

### 2.2. RAPID Algorithm of Guided Wave Tomography

For conventional UT inspection, wave signal change is a key factor in evaluating safety conditions. Especially, in the case of GW inspection, wave propagating motion is affected by the waveguide status and geometric variation. The well-known verification techniques are time-of-flight and amplitude analysis (A, B, or C-scan), which are the most common methods to reconstruct images from the elastic wave signal. For the sake of improving detection capability, a large number of transducers should be installed at high-risk areas of the structure. However, in the case of inaccessible conditions or limited number of sensors, increasing the number of transducers is not practical. In order to overcome these problems, the RAPID algorithm is an alternative way to evaluate the facilities [9]. The RAPID algorithm is based on a signal difference coefficient (SDC) in accordance with compatibility with embedded sensor applications as shown in Figure 2. The basic principle of SDC, Equation (5), compares the damaged and non-damaged statuses which are related with Cross-correlation function. The time variable of $t_{0}$ is the direct arrival time for each transducer pair. Here, $\mu_{x}$ and $\mu_{y}$ are the mean of the respective data set of X and Y, respectively. In this case, the data set X is the reference data and Y is each new set of data recorded after a period of the service time. If the signals are identical to the no-defect condition, the SDC is zero and, if the signals are completely out of phase, the SDC will achieve a maximum value of one. After the SDC value for all sensor pairs are calculated, the next step of the RAPID algorithm is reconstruction:(5)$${\mathrm{SDC}}_{ij}=1-\left|\frac{\int_{t_{0}}^{t_{0}+\Delta T}\left[x_{ij}(t)-\mu \right]\;\left[y_{ij}(t)-\mu \right]dt}{\sqrt{\int_{t_{0}}^{t_{0}+\Delta T}{\left[x_{ij}(t)-\mu \right]}^{2}dt\int_{t_{0}}^{t_{0}+\Delta T}{\left[y_{ij}(t)-\mu \right]}^{2}dt}}\right|.$$

Tomographic visualization is accomplished by spatially distributing each SDC value in an elliptical pattern. A parameter β is defined to control the size of the ellipse. The amplitude tapers from its maximum value along the line connecting the ellipse foci to zero on its periphery. Note that parameter β is defined as the shape factor that controls the size of the elliptical distribution:(6)$$s(x,y,x_{TK},y_{TK},x_{RK},y_{RK})=\frac{\beta -R(x,y,x_{TK},y_{TK},x_{RK},y_{RK})}{1-\beta }\;\mathrm{for}\beta >R(x,y,x_{TK},y_{TK},x_{RK},y_{RK}),\;s(x,y,x_{TK},y_{TK},x_{RK},y_{RK})=0,\;\mathrm{otherwise},$$
where $R(x,y,x_{TK},y_{TK},x_{RK},y_{RK})$ is the ratio of the sum of distances of the point (x,y) to the transmitter $\left(x_{TK},y_{TK}\right)$ and receiver $\left(x_{RK,}y_{RK}\right)$ to the distance between the transmitter and receiver and is mathematically stated as
(7)$$R(x,y,x_{TK},y_{TK},x_{RK},y_{RK})=\frac{\sqrt{{\left(x-x_{TK}\right)}^{2}+{\left(y-y_{TK}\right)}^{2}}+\sqrt{{\left(x-x_{RK}\right)}^{2}+{\left(y-y_{RK}\right)}^{2}}}{\sqrt{{\left(x_{TK}-x_{RK}\right)}^{2}+{\left(y_{TK}-y_{RK}\right)}^{2}}}.$$

Finally, the tomographic image amplitude at each pixel is calculated as the linear summation of the location probabilities from each transmitter and receiver $s(x,y,x_{TK},y_{TK},x_{RK},y_{RK})$ pair with the total number of transmitter–receiver pairs given by M and is stated as
(8)$$P(x,y)=\sum_{k=1}^{M}p_{k}(x,y)=\sum_{k=1}^{M}\frac{SDC}{\beta -1}\left\{\beta -R(x,y,x_{TK},y_{TK},x_{RK},y_{RK}\right\},1\leq R\leq \beta .$$

## 3. Experimental Setup and Specimen Information

In this section, to verify the suitable GW mode and image tomographic visualization, the Korea hydro and nuclear power company (KHNP) has been consulted to design the CLP mock-up specimen. As mentioned in the introduction, there are three types of wave modes: symmetric, antisymmetric, and Rayleigh surface wave. An experimental setup has been designed considering the supply of stable power flux and a sufficient level of energy. 

### 3.1. Specimen Information

The CLP mock-up specimen is a type of carbon-steel plate, SA516 GR 60, the thickness of which is 6 mm. The dimensions of the plate (width and height) are 2500 × 1400 mm, as shown in Figure 3. Information on the defects is given in Table 1. In order to view the image precisely, the specimen is divided into eight parts as illustrated in Figure 4. As a part of the CLP mock-up specimen, an inspection area of 600 × 600 mm is selected with a propagation distance in the range of 210 to 848 mm.

For the purpose of collecting GW signals, the transducer array locations are selected as shown in Figure 5. From the viewpoint of basic SHM applications, robotics or built-in sensors should be discussed. Therefore, in light of the attenuation of GW propagation and the realization of a maximum-sized tomography, a total of 16 transducer location positions are selected and 160 wave signals are processed by each local part. At the end of these processes, a complete image is reconstructed by combining each tomographic image part. 

### 3.2. Experimental Setup

To realize a GW-based SHM application, the experimental setup is designed to generate sufficient energy levels of the GW propagated on the plate. A high-power tone burst system (RPR-4000, RITEC, Warwick, RI, USA) is connected with an oscilloscope (Wave runner 640zi, Teledyne Lecroy, Rockland Country, NY, USA) and a laptop computer for data analysis. The experimental setup is systemized to generate Lamb waves as shown in Figure 6.

In order to transmit and receive the GW signal, a piezoelectric transducer (PZT) is employed to the CLP mock-up specimen. The transducer is made from PZT and its center frequency is 1.0 MHz (Olympus, standard square shape, 0.5 × 1.0 in^2^). A type of contact based experiment is performed using acoustic couplant (SONOTECH, ultragel II). The transducer is fixed by pressure on the wedge with 186 kN/m^2^ by a manual method. The complete wave propagation network of each part is illustrated in Figure 5. In this study, contact-based wave generation is treated to evaluate the CLP mock-up specimen. In the case of contact-based GW generation, the exact incidence angle is required to be taken into account for proper wave propagation. In the case of symmetric and antisymmetric Lamb wave mode generation, the incidence angle is 85°, 34°, 27°, 27°, 50° and 30°, respectively. At the beginning of the experiment, GW signals are acquired. The symmetric (S0 2.0 MHz·mm, S1 3.5 MHz·mm and S0 3.0 MHz·mm), antisymmetric (A0 1.0 MHz·mm, A1 3.0 MHz·mm and A1 3.6 MHz·mm) wave modes are successfully generated as shown in Figure 7.

## 4. Results and Discussion

To verify the wall-thinning defects and the feasibility of GW-based SHM tomography, the RAPID algorithm, as discussed in Section 2.2, is applied to the CLP mock-up specimen. Tomographic image analysis is an alternative technique to consider appropriate GW mode selection from a practical viewpoint. Evidence shows that the RAPID visualization result for the same wall-thinning case is undertaken by suitable GW mode selection.

In the case of the anti-symmetric mode, A1 1.0 MHz·mm scarcely appears on the oscilloscope and its amplitude is too small to generate. The mode has a disadvantage of attenuation with respect to propagation distance. The result of the A1 3.0 MHz·mm case represents a larger amplitude than A1 3.6 MHz·mm. However, the A1 3.0 MHz·mm case has too many wave modes that are superpositioned, relatively. An A1 3.6 MHz·mm signal is identified as the dominant wave mode. In the case of symmetric mode, S1 3.5 MHz·mm, similar to the A1 1.0 MHz·mm case, resulted in too small of an amplitude. S0 2.0 MHz·mm appears to be a relatively sufficient amplitude manner. However, the wave mode also superpositioned different modes that are not available to evaluate defects. Therefore, S0 3.0 MHz·mm and A1 3.6 MHz·mm are considered as the most suitable wave modes to evaluate the CLP. In order to figure out the characteristic of wave fluctuation, the wave structure curve is illustrated in Figure 8. The units of Figure 8 are normalized values illustrating displacement and power distribution [2]. As shown in the wave structures, both antisymmetric and symmetric Lamb wave modes can be regarded as proper GW modes for the wall-thinning failure case of which power flux and displacement are distributed on the surface sufficiently. In contrast, the CLP mock-up specimen is relatively thicker than the ones used in other research works and the energy loss is a more sensitive factor for applying Lamb wave.

### 4.1. Finite Element Analysis for Suitable Mode Verification

The selection of an appropriate mode is essential to realize GW-based SHM. In order to verify the availability of the Lamb wave propagation, FEA is applied preferentially. For waveguide modeling and analysis, the commercial software, ABAQUS is explicitly used to simulate Lamb wave scattering, and a tone-burst windowing signal is generated by the MATLAB coding process. For generating Lamb wave signals, S0 (3.0 MHz·mm) and A1 (3.6 MHz·mm) are simulated by ABAQUS explicit modeling. Five wave form cycles of 500 kHz and 600 kHz wave sinusoidal signal tabular data are generated by MATLAB for S0 and A1 modes, respectively; this is shown in Figure 9. To reduce spectral leakage, the Hann window function is applied to each of the sinusoidal signals in Equation (9), where *n* is the number of digits and *N* is total digit number. The window size is set by total 50 digits on five cycles, composed 10 digits per cycle:(9)$$w(n)=0.5\left(1-\cos\left(2\pi \frac{n}{N}\right)\right),\;0\leq n\leq N.$$

As mentioned in Section 4, the material is a type of carbon-steel; its properties are as follows: Young’s modulus E = 200 GPa, density ρ = 7.87 g/cm^3^, and Poisson’s ratio ν = 0.29. The phase velocities of the S0 and A1 modes are 3520 m/s and 5510 m/s, respectively, the wavelength of which is calculated as 7 mm and 9 mm, respectively. 

The model of the plate is designed for 2D analysis and its dimensions (width and height) are 600 × 6 mm, as shown in Figure 10. The size of a mesh relies on the wavelength such that the mesh size (S0 = 0.7 mm, A1 = 0.9 mm) is 1/10th of the wavelength. Both of the approximated S0 and A1 modes’ mesh sizes are 0.667 mm and 0.856 mm, respectively, as illustrated in Figure 11. The mesh is of a quadrangle type and the viscosity and damping effects are neglected. The set number of mesh elements is 7722 and 4662 for the S0 and A1 modes, respectively. 

From the results of the FEA simulation, the Lamb wave mode signal is successfully collected as shown in Figure 12. In the case of A1 mode, many GW modes superpose and the dominant mode is not clearly plotted. In contrast, the S0 mode result indicates that a fewer number of modes are superposed than in the case of the A1 mode. In addition, a comparison between the A1 and S0 motions shown in Figure 13 indicates that A1 wave patterns are relatively less well distinguished. The wave form of the S0 mode is more clearly expressed than that of the A1 mode. The wave power of each mode is analyzed and the result shows that the S0 mode (2.352 × 10^−13^) is four times larger than the A1 mode (5.76 × 10^−14^).

Additionally, wave energy variation with respect to the wall-thinning depth is also studied to verify the selection of an appropriate mode. The energy properties should be a key factor for the availability of the GW based on RAPID tomography. The wave power function (P), depending on the wave velocity, frequency, and amplitude, is defined as [2]:(10)$$P=\frac{1}{2}\rho c\int_{t_{0}}^{t_{0}+\Delta t}A{\left(x\right)}^{2}\omega^{2}dt,\;\rho ,\;c\;\mathrm{and}\;\omega \;\mathrm{are}\;\mathrm{constant}.\;∴\;P\propto \sum_{n=1}^{N}A{\left(n\right)}^{2},$$
where ρ, c, ω, and A are the density, wave velocity, angular frequency, and amplitude, respectively. As the wave energy is calculated under the same wave velocity and frequency status, the only interesting factor is the wave amplitude. Accordingly, the square values of the amplitudes are required to compare energy variation with respect to the wall-thinning depth. The wall-thinning defect is reduced from 0 to 4 mm in the steps of 1 mm; thus, four steps are applied to CLP mock-up modeling. As an exception, from the case of the 1 mm depth verification using the A1 mode, the result shows that the wave energy is increased in comparison to the no-defect case that is related to mode conversion and the characteristics of stress–strain energy distribution on the A1 mode, as shown in Figure 14. This phenomenon should be treated as future research. With the overall procedures using FEA, the S0 mode is regarded as the most suitable wave mode for the CLP mock-up evaluation. 

### 4.2. Wall-Thinning Verification for Lamb Wave Modes

In the previous Section 4.1, from the result of a comparison of wave energy using FEA, S0 mode is regarded as the proper mode that can offer reasonable wave energy changes with respect to variation in the wall-thinning depth. The verification method, similar to FEA, involving wave shape and wave energy alteration in accordance with the wall-thinning status, is performed for the CLP mock-up specimen. For the step of a wall-thinning defect of 1 mm that is reduced from 0 to 2 mm, three steps are applied to CLP mock-up. The defect numbers #5 and #9 are located in the same area, for which the wall-thinning depths are 2 mm and 1 mm, respectively, as explained in Table 1. The shape information of defects is illustrated in Figure 15. 

As the wave is transmitted in the wall-depth region, the waveform is changed with the wall-thinning depth. In the case of no defect condition, maximum amplitudes are 119 mV and 80 mV, respectively. In the case of 1 mm wall-thinning depth, the S0 mode Lamb wave amplitude, which is related to energy, is decreased and the mode conversion effect is observed as well. As clarified by group velocity analysis, the wave mode propagated from S0 3.0 MHz·mm (maximum amplitude: 50 mV) is a 2.5 MHz·mm Lamb wave mode (maximum amplitude: 45 mV) that can appear at 5 mm thickness under 500 kHz wave signal excitation. In contrast, for the case of A1 mode, in the same manner as the FEA case, the wave amplitude is enhanced as it propagates through the defect area of which the maximum amplitude is 118 mV. Furthermore, the group velocity is shifted to 2670 m/s, and this mode is regarded as A1 4.0 MHz·mm, S1 2.9 MHz·mm and Rayleigh surface wave. In the case of 2 mm defect depth case, both modes are monitored decreasing amplitudes. The maximum amplitudes of A1 3.6 MHz·mm and S0 3.0 MHz·mm are 31 mV and 48 mV, respectively. These observations provide a basis to further explore wall-thinning verification in future research.

In the case of 2 mm wall-thinning depth, both S0 and A1 modes’ amplitudes are decreased as shown in Figure 16. Especially, for the case of A1 mode, the amplitude is more rapidly reduced compared with the S0 mode case, which can be inferred from the wave structure shown in Figure 8. The wave structure analysis for A1 mode reveals that the out-of-plane displacement variation is more sensitive than that for the S0 mode. The result of wave shape analysis indicates that the S0 mode is more suitable than the A1 mode because the amplitude is well-attenuated in accordance with the wall-thinning depth. In the previous section, three kinds of GW modes, S0, A1 and Rayleigh surface modes, have been discussed from the viewpoint of wave energy variation and GW generation. In this section, the optimum GW mode is determined through tomographic image reconstruction.

### 4.3. Tomography Comparison Study for Suitable Guided Wave Modes

The RAPID tomographic algorithm is a key technique to determine the suitable mode. Tomographic images, obtained from quantitative GW wave signal analysis, not only allow straightforward detection of integrity but also facilitate selection of the suitable GW mode. Each GW mode is verified using the reconstruction images obtained from the RAPID algorithm for the case of Part 1 on the CLP mock-up specimen. 

Part 1 of CLP mock-up is used to reconstruct the RAPID algorithm image, for which the wave propagation network and defect locations are shown in Figure 17; the defect information is also provided in Table 1. The A1 mode is reconstructed as shown in Figure 18. The A1 mode tomography does not well represent the size and location of defects. As mentioned in the previous section, the S0 mode is expected to a suitable mode for GW based SHM. The RAPID tomography image, using S0 Lamb wave mode, better represents the defect information compared with the other GW modes shown in Figure 18. From the overall result of mode analysis, the most proper GW mode is determined to be S0 3.0 MHz·mm.

### 4.4. CLP Mock-up Reconstruction Using RAPID Algorithm Tomography

The tomographic reconstruction images, based on the RAPID algorithm, are illustrated for the CLP mock specimen. In Section 4.1, the CLP mock up is split into eight parts considering the limitation of reasonable GW wave propagation distance and for enhancing the accuracy of the tomographic image. The most suitable GW mode, S0 3.0 MHz·mm, is applied to evaluate the status of each part. The Lamb wave inspection is undertaken five times for each propagation way. Eight tomographic parts images are quantitatively processed by 160 datasets for each part. Subsequently, data for 1280 wave signals are analyzed for visualization. The tomographic images for all parts are illustrated in Figure 19. Parts 1, 2, 5, 7 and 8 are acceptable images, showing the defects’ shape and location same as those on the CLP mock-up. In cases of parts 3, 6 and 7, defects, located near the transducer position, are visualized with good agreement of location.

In contrast, shapes of the defects are not imaged relatively well. Those results are likelihood conjectured that come from the position of transducer, narrowed location with each other, and defects that can cause significant effects on the acquisition of signal, such as refraction, diffraction and high attenuation. In addition, from the practical viewpoint, the variable shape parameter β in the RAPID algorithm is not applied to the results, which is estimated to affect the shape of the reconstructed images. Data obtained in previous studies using the variable shape parameter are used to obtain exquisite and enhanced reconstruction images [21,23].

The tomographic visualization result of the CLP mock-up specimen, combining all parts of the reconstruction images, is shown in Figure 20. It is noticed that the combined image well matches the defect information. Moreover, the location information is successfully determined and the defects shape is also illustrated, which are acceptable outcomes from GW based SHM. Finally, the significant result from GW based SHM is that the tomographic image reconstruction, based on the RAPID algorithm, is successfully illustrated using the symmetric 3.0 MHz·mm Lamb wave mode. 

## 5. Conclusions

In this study, the GW based SHM technique is used to evaluate CLP, which is a high-risk facility of nuclear power plants, for enhancing performance and visualization. The RAPID tomographic algorithm is applied to reconstruct specimen images. In order to evaluate the CLP mock-up specimen with GW based SHM, appropriate mode selection is required for generation of the Lamb wave mode. Three kinds of GW mode, S0 3.0 MHz·mm, A1 3.6 MHz·mm and Rayleigh surface wave modes, are studied to determine the suitable mode. Those wave modes are verified through FEA, experiment and visualization. S0 and A1 modes are considered preferable from the analysis of wave structures. From the viewpoint of surface sensitivity, the A1 mode is expected to be the most suitable for the wall-thinning defect case. However, it appears that the S0 mode, which is the dominant mode of longitudinal waves, is more suitable than the A1 mode due to better propagating performance for thicker specimens. At the beginning of mode selection, FEA is performed to verify mode generation and energy level variation. For the Lamb wave mode cases, the A1 mode wave propagation shows that ambiguous modes are generated with mode superposition; the S0 mode shows relatively good dominance. For wall-thinning depth verification, the S0 mode energy level is decreased in accordance with the increase in the defects’ depth. On the other hand, for the A1 mode, the wave energy level is exceptionally elevated in the 1 mm depth case. From the FEA result, the S0 mode is the most suitable for application in GW based SHM. Experimental studies are also undertaken. Similar to the FEA result, the A1 mode energy level is elevated in the 1 mm depth case and change in group velocity is also observed, which means that the A1 mode has a likelihood that changed different modes. To perform image reconstruction, the specimen is divided into eight regions to enhance the completeness. The investigation of wave modes using visualization techniques, based on the RAPID algorithm, is performed on CLP part no. 1. The S0 mode is illustrated to generate the most well matched images from the tomography. From the overall results of GW mode verification, the S0 3.0 MHz·mm is used to reconstruct quantitative tomographic images for the CLP mock-up specimen. The location information is well identified and the defects shape is also well illustrated from GW based SHM. Finally, the significant result from GW based SHM is that the tomographic image reconstruction, based on the RAPID algorithm, is successfully illustrated using the symmetric 3.0 MHz·mm Lamb wave mode. In future work, the RAPID tomographic algorithm involving the variable shape parameter technique [21,23] should be used to improve visualization of the shape of defects for the CLP facility. In addition, with the application of omnidirectional transducer, an annular type of transmitter–receiver [28,30], this evaluation technique is expected to reinforce the reliability and feasibility to monitor CLP.

## Figures and Tables

**Figure 1 sensors-19-02819-f001:**
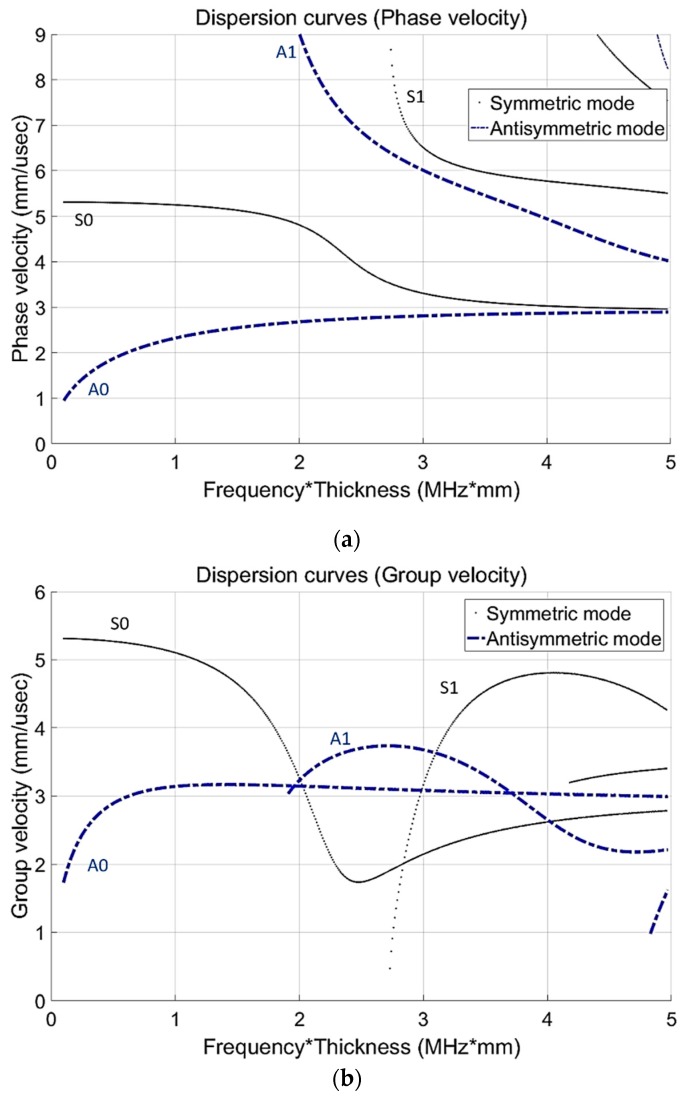
Dispersion curve for CLP mock-up specimen. (**a**) dispersion curve for phase velocity (**b**) dispersion curve for group velocity.

**Figure 2 sensors-19-02819-f002:**
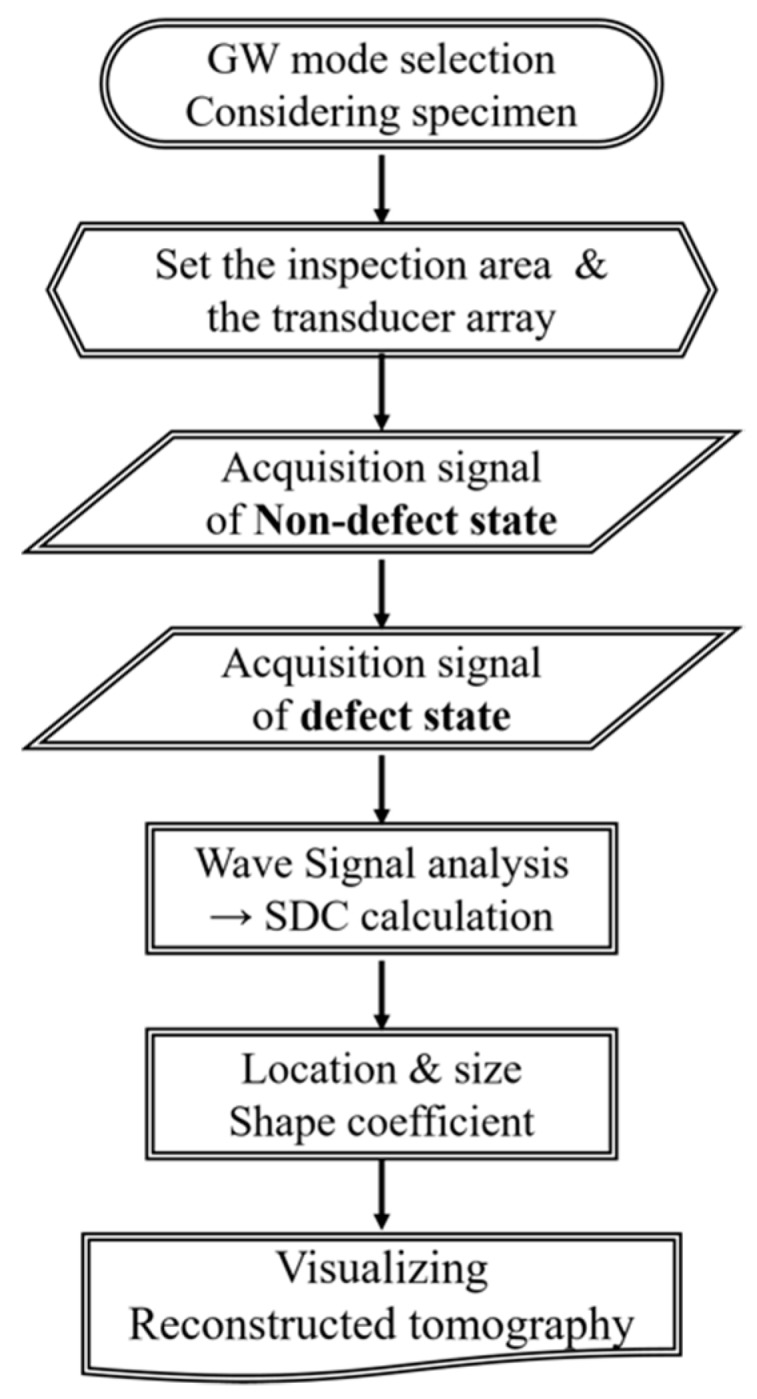
Basic concept of image reconstruction using the RAPID algorithm.

**Figure 3 sensors-19-02819-f003:**
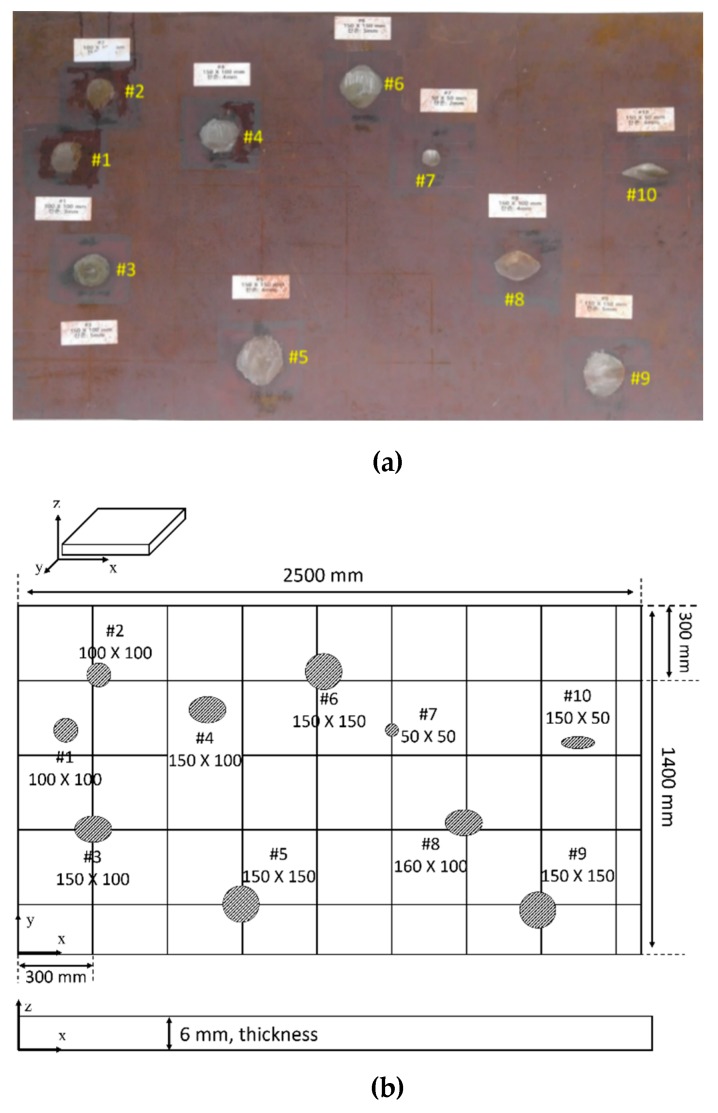
Schematic of CLP mock-up specimen (**a**) real positions of defect locations; (**b**) size information of defects on the CLP mock-up plate.

**Figure 4 sensors-19-02819-f004:**
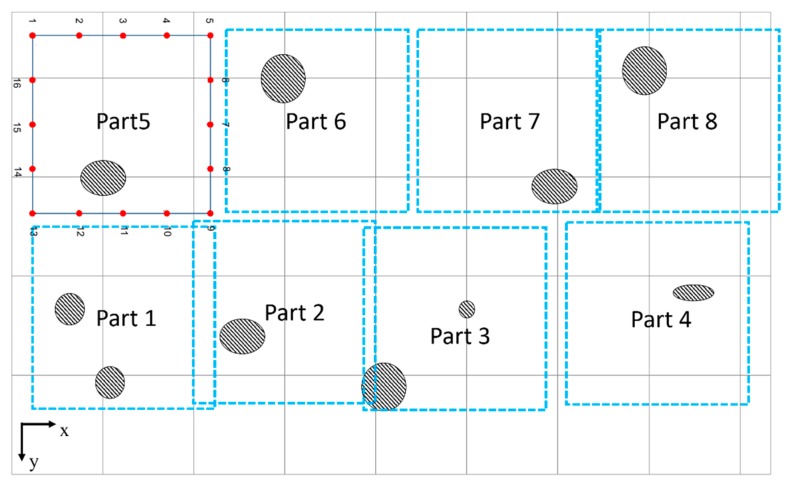
Partition information of the CLP mock-up specimen.

**Figure 5 sensors-19-02819-f005:**
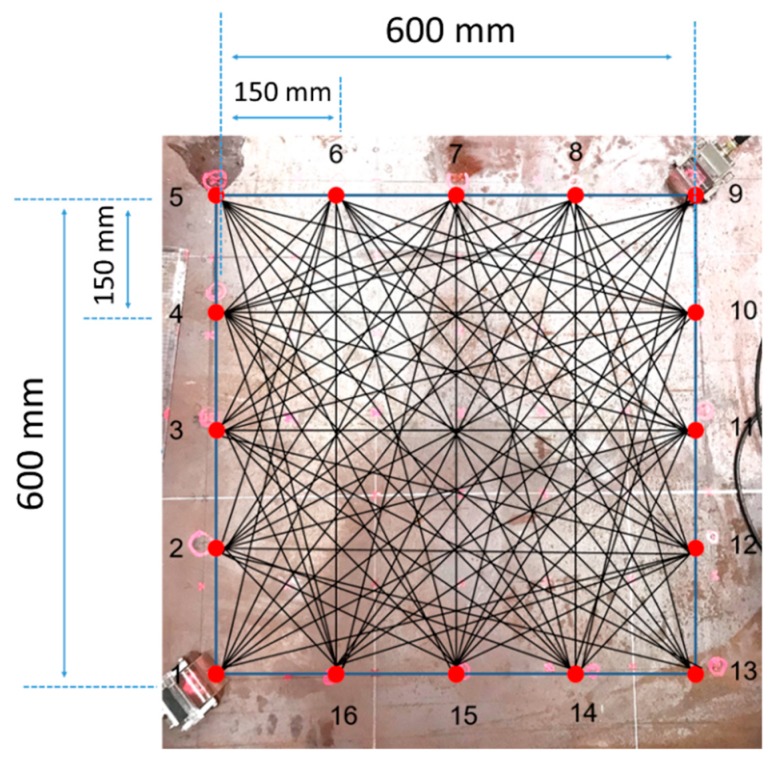
Schematic of part size and transducer location on the specimen (indicated as red circles).

**Figure 6 sensors-19-02819-f006:**
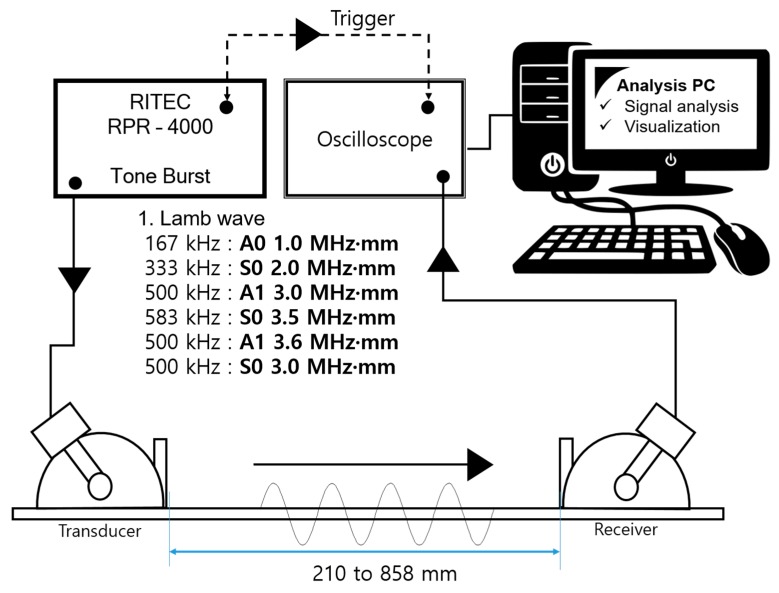
Experimental setup for generating GWs.

**Figure 7 sensors-19-02819-f007:**
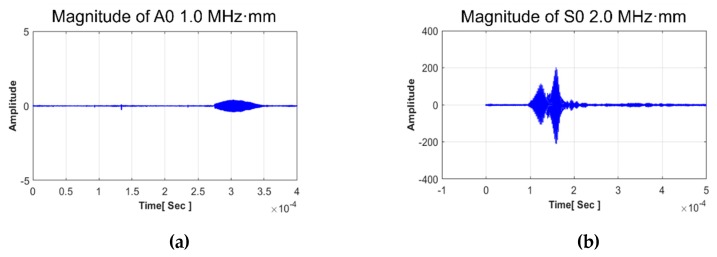
Magnitude of GW signal from transmitter–receiver pair under 400 mm propagation distance (**a**) A0 1.0 MHz·mm; (**b**) S0 2.0 MHz·mm; (**c**) A1 3.0 MHz·mm; (**d**) S1 3.5 MHz·mm; (**e**) A1 3.6 MHz·mm; (**f**) S0 3.0 MHz·mm.

**Figure 8 sensors-19-02819-f008:**
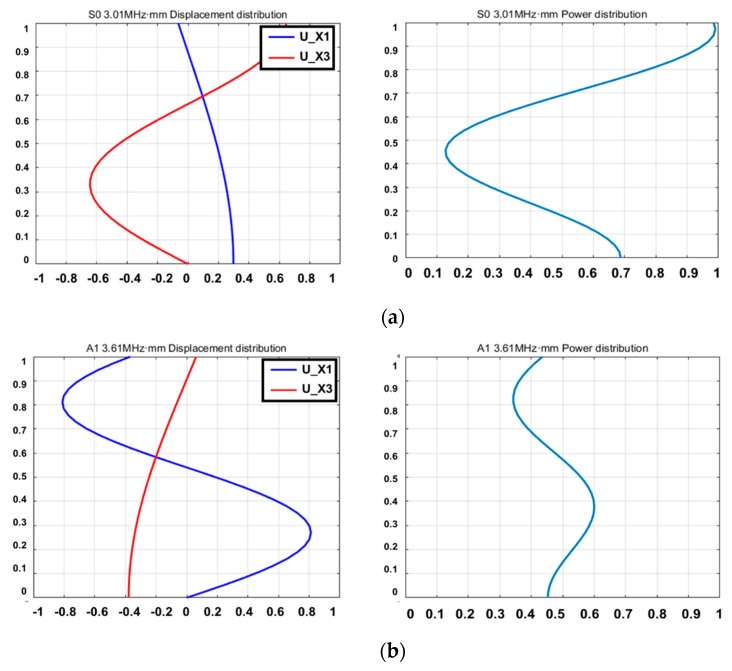
Wave structure analysis of Lamb wave modes; (**a**) S0 3.0 MHz·mm; (**b**) A1 3.6 MHz·mm.

**Figure 9 sensors-19-02819-f009:**
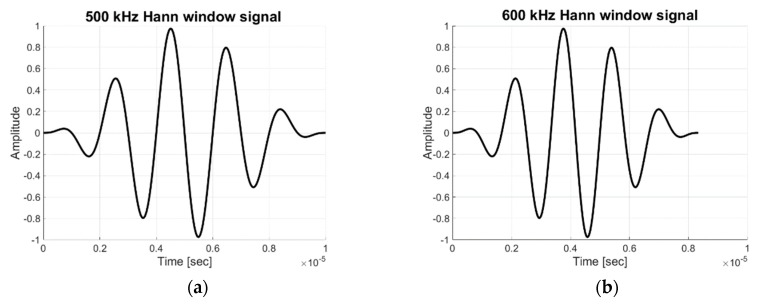
Hann window signal for excitation of wave signal (**a**) 500 kHz for S0 mode; (**b**) 600 kHz for A1 mode.

**Figure 10 sensors-19-02819-f010:**
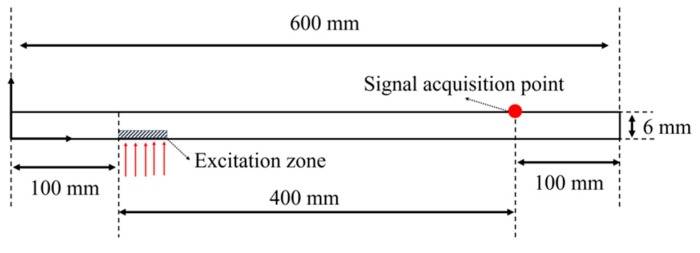
CLP mock-up modeling specification for FEA simulation.

**Figure 11 sensors-19-02819-f011:**
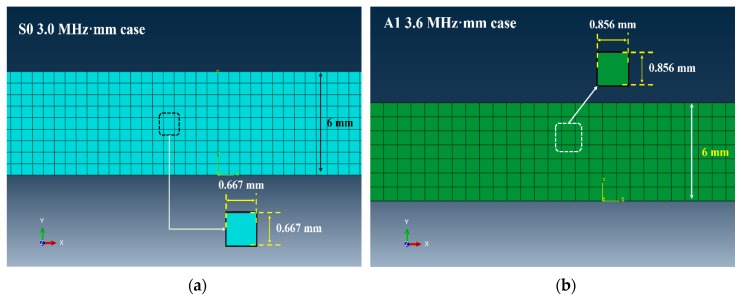
Geometric information of mesh for generating Lamb wave (**a**) 500 kHz for S0 mode; (**b**) 600 kHz for A1 mode.

**Figure 12 sensors-19-02819-f012:**
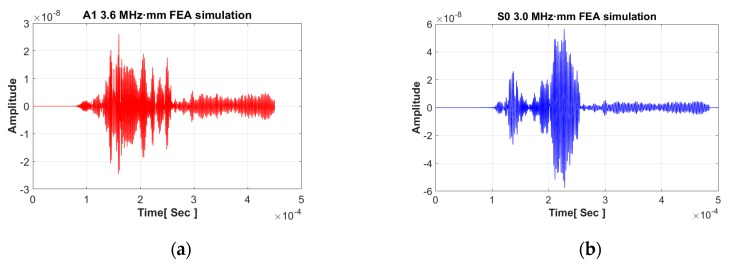
Magnitude of GW signal from ABAQUS FEA simulation under 400 mm propagation distance (**a**) A1 3.6 MHz·mm; (**b**) S0 3.0 MHz·mm.

**Figure 13 sensors-19-02819-f013:**
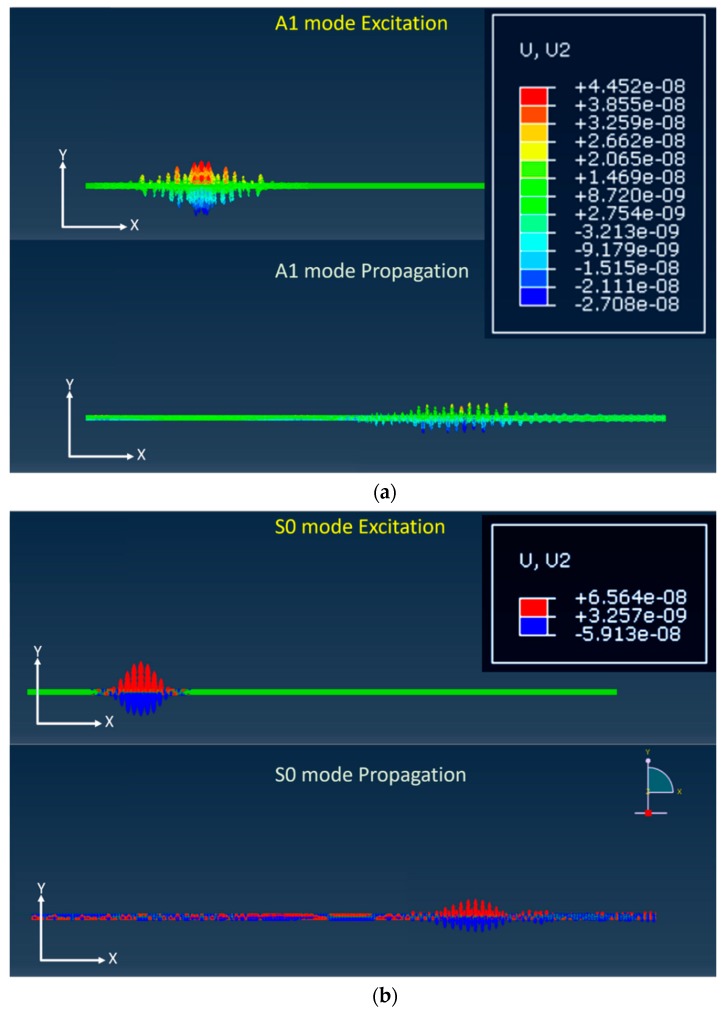
Propagation motions of Lamb waves from ABAQUS FEA simulation under 400 mm propagation distance (**a**) A1 3.6 MHz·mm; (**b**) S0 3.0 MHz·mm.

**Figure 14 sensors-19-02819-f014:**
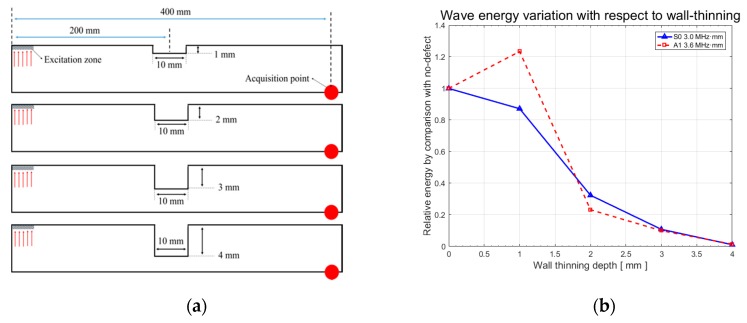
Lamb wave energy variation with respect to wall-thinning depth (**a**) modeling of wall-thinning defects for FEA; (**b**) energy variation results in comparison with a no-defect condition.

**Figure 15 sensors-19-02819-f015:**
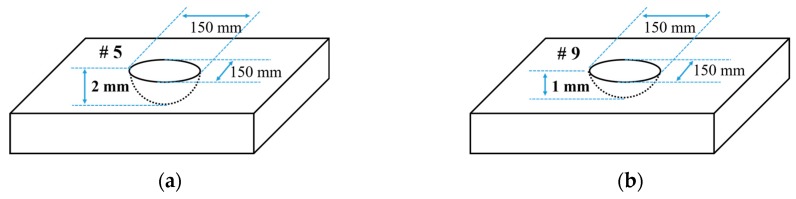
Wall-thinning defects information for experimental verification: (**a**) defect No. 5 with wall-thinning depth = 2 mm; (**b**) defect No. 9 with wall-thinning depth = 1 mm.

**Figure 16 sensors-19-02819-f016:**
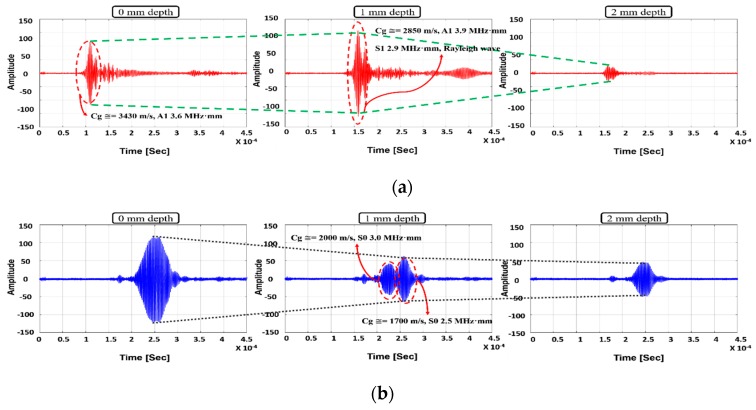
Experimental amplitude and wave mode variation of Lamb wave modes with respect to wall-thinning status: (**a**) A1 3.6 MHz·mm; (**b**) S0 3.0 MHz·mm.

**Figure 17 sensors-19-02819-f017:**
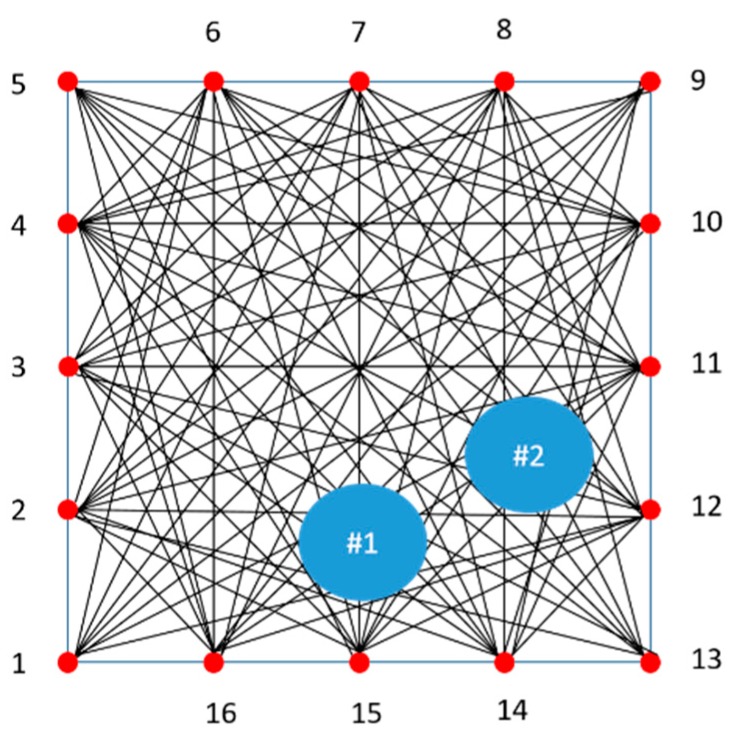
Wave propagation network and defect locations of Part 1 for verifying suitable mode selection by image reconstruction techniques.

**Figure 18 sensors-19-02819-f018:**
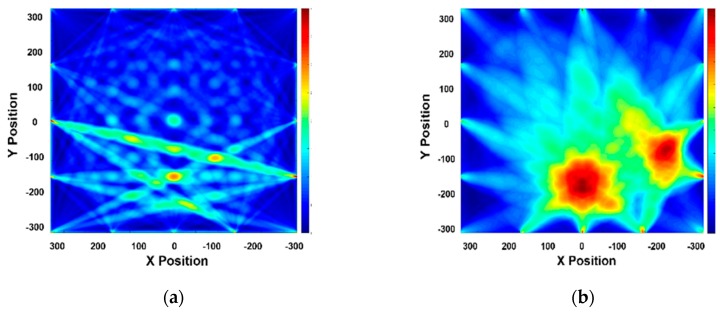
RAPID tomographic results for GW modes: (**a**) A1 3.6 MHz·mm case; (**b**) S0 3.0 MHz·mm case.

**Figure 19 sensors-19-02819-f019:**
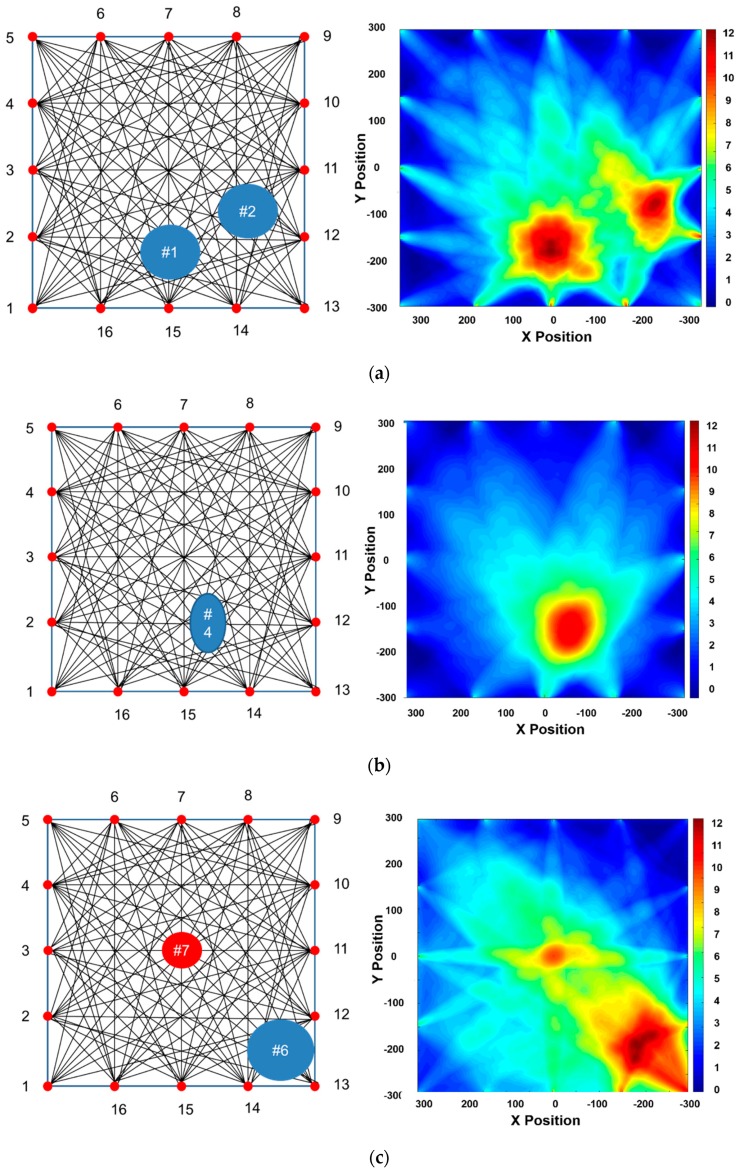
Overall results of tomographic images for CLP mock-up specimen: (**a**) part 1; (**b**) part 2; (**c**) part 3; (**d**) part 4; (**e**) part 5; (**f**) part 6; (**g**) part 7; (**h**) part 8.

**Figure 20 sensors-19-02819-f020:**
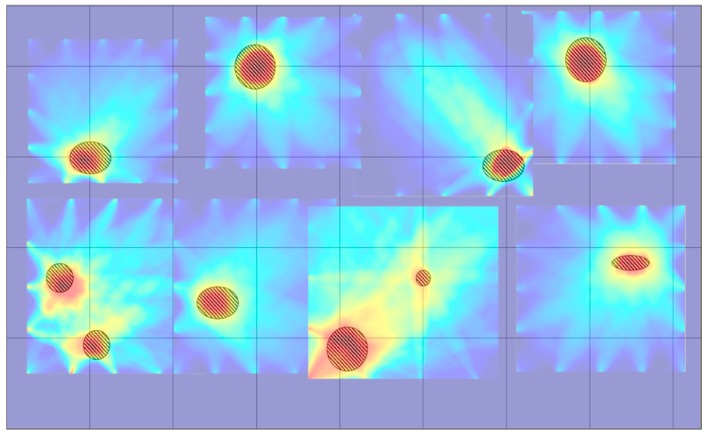
Tomography result of CLP mock-up specimen: comparison of defects information and tomographic image of CLP mock-up specimen.

**Table 1 sensors-19-02819-t001:** Geometrical information of defects on the CLP mock-up specimen.

Defects #	Thickness (mm)	Size (mm)
1	3.1	100 × 100
2	5.3	100 × 100
3	4.7	150 × 100
4	5.9	150 × 100
5	4.0	150 × 150
6	5.9	150 × 150
7	2.0	50 × 50
8	4.0	160 × 100
9	5.0	150 × 150
10	5.9	150 × 50

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
