# Peer review of "An Investigation on a Quantitative Tomographic SHM Technique for a Containment Liner Plate in a Nuclear Power Plant with Guided Wave Mode Selection"

_sensors, 2019, doi:10.3390/s19122819_

Reviewer 1 Report

The paper presents interesting numerical simuations and experimental results on defect detection with ultrasonic guided waves. However, the manuscript promises more than it can fulfill. In the abstract it promises: "The proposed SHM technique relies on sensors and therefore, it can be placed on the structure permanently and can monitor either passively or actively." In fact, Rayleigh and Lamb waves have a distinct propagation direction. The transmitter and receiver have to point to each other. The transducers cannot be in a fixed position for all measurements. Example: for transmission 1 -> 6 the transducer 1 has a different orientation than for the transmission 1 -> 7, or 1 -> 8 etc. Hence the proposed measurement cannot be done with fixed transducers.

p 2, line 64: Finite Element Analysis (FEA)

p 3, Fig. 1 and line 110: write consistently the index as a subscript

p 4, line 127: give clear justification why these specific values were chosen

p 4, eq 8: cp instead of cR

p 4, first sentence after eq. 8: Give here values for specimen thickness in this work and references for "other Lamb wave inspection"

p 5, Fig. 3: in right hand figure: "mode" is partly covered

p 5, Fig. 4: give units; give formula of power

p 6ff, eq 10-13: give references

p 7, eq 13: this number is given twice

p 8, line 204: give here formula how the power is calculated

p 8, Fig. 8: "mm" of thickness is missing

p 8, Fig. 9: labeling of axes is too small. Time scaling is not visible

p 9, Fig. 10: Figure labeling too small. What is x-axis, what is y-axis?. Labels of color bar is not readable

p 9, eq 15: give reference

p 10, Fig. 12b: #8 is 160x100 according to Table 1

p 11, line 267f: give manufacturer, city, country

p 12, line 271: how large is the Rayleigh wavelength?

p 12, line 277: Give manufacturer, type, frequency of used transducers. Give information on coupling agent and how the transducers were pressed onto the surface (manually or with mechanized/robotized system).

p 12, Fig. 15: separation of transducers: delete digits after comma

p 12, Fig. 16: labeling fonts are too small

p 13, Fig. 17: This defect shape information should be given in sect. 4.1, not here

p 13, Fig. 18: Label fonts are too small

p 14, Fig. 19: Transmitter and receiver have to point exactly to each other (wedges!). How is this adjustment achieved? How long took such a complete data acquisition for one tomogram

p 14, Fig. 20: label fonts are too small

p 15, Fig. 21: label fonts are too small

Author Response

Dear, Reviewer. 

I appreciate your advice and comment. 

We tried to fix what you ask to correct and edit part.

Most of the parts of the manuscript are fixed and edited.

It is really helpful and considerable works for this research work.

Probably, some edited comments and supplements might not be satisfied with your opinion.

Regardless of this problem, please give us good pieces of advice and opinion for improvement.

Thank for reviewing the manuscript.

Sincerely yours

Author: Yonghee Lee, Younho Cho.

Reviewer 2 Report

This paper studies the tomography for wall-thinning depth verification using ultrasonic guided wave. The authors applied the tomography method to evaluate the container liner plate. An innovative contribution of this paper either in the algorithm or in the evaluation method should be emphasized in the introduction section. The theoretic background of lamb wave and introduction of RAPID algorithm are suggested to be shortened since it is not the major contribution of this paper.  The application of the proposed approach in the structural health monitoring of plate structures in the energy industry can be added to enrich the conclusion sections.  

Author Response

(The authors gave the same response as above.)

Reviewer 3 Report

find attached file.

Author Response

(The authors gave the same response as above.)

Reviewer 4 Report

The authors utilize current well-established methods such as a RAPID algorithm for inspection of isotropic plates. The aim is tomography of wall-thinning. The topic is important and the author's main focus is the monitoring of containment liner plate in a nuclear power plant. Despite the fact that the authors mostly use existing methods, their experimental campaign is quite broad and the obtained results may be of interests of potential readers. Unfortunately, the paper is chaotic and not enough clear for potential readers. It needs improvements in many places.

The detailed remarks are below:

The title is inappropriate. The authors are able to visualize the location and size of defects in the form of loss of material but the estimation of defect depth is only barely addressed. In fact, it can be very difficult to achieve based on experimental signals.

Section 2.1 can be shortened because it repeats only well-known theory.

Page 4 “As shown in the wave structures, the antisymmetric Lamb wave might be expected to proper GW mode for the wall-thinning failure case of which power flux and displacement distributed on the surface” This sentence is unclear and should be rewritten.

It is not clear why the authors recall eq. 9.

It is not clear why the authors selected particular excitation frequencies.

Page 7, line 179 “tablet data “ It should be rather “tabular data”

What is the reasoning behind showing the sinusoidal signal in Fig. 6? In my opinion, it should be removed.

How the excitation was realized in simulations? Forces were applied?

From dispersion curves, it is clear that at the analysed frequencies 4 modes propagate simultaneously. Therefore, it is not clear what mean captions in Fig. 9 related to A1 and S0 mode only. Was there any special technique to excite particular modes?

Page 9, Eq. 15 What is the window size?

Page 9, line 227, “the S0 mode is regarded as the most suitable wave mode for the CLP mock-up evaluation” Why?

Table 1, Is it plate thickness or defect depth?

Figure 12, What is transducers placement with respect to the plate and defects? Are the transducers located at the boundaries of each part shown in Fig. 13? It seems that the dimensions of each part should be 600x600 mm but in Figure 13 parts seems to have various dimensions. Please add transducer locations to Fig. 13.

Fig. 15 It does not make sense to show the same figure twice. The only difference is the excitation frequency.

Page 13, line 306 “The step of wall-thinning defect 1 mm that is reduced from 0 to 2 mm, 3 steps are applied to CLP “ This sentence must be rewritten.

Figure 19, Defect locations are not compatible with defects shown in Fig. 13.

Page 15, line 350 “As mentioned in the previous section, the S0 mode is expected to a suitable mode for GW based SHM” It should be rewritten.

The verb is missing in the first sentence of acknowledgements.

Author Response

(The authors gave the same response as above.)

Reviewer 5 Report

The paper deals with the use of RAPID technique for thick specimens from the nuclear power plant. 

The paper provides experimental and numerical validation but considerable work needs to be carried out on the manuscript before the paper will be of quality expected from an international peer- reviewed journal. The major concerns the reviewer has are:

1) The highlights and the novelty of the paper has not been clearly highlighted

2) The reviewed literature is not in detail enough. Very few papers from the last 5 years have been cited. The reviewer will encourage the literature review on the use of tomography based techniques as well as SHM techniques in nuclear power plants.

3) The paper structure is cluttered with several significant details not provided and largely trivial information and background provided. The background about GW is well established including Figure 2 and can be omitted. On the other hand a photo or a schematic of the application proposed in the literature will help the understanding. Also the relation of the mock-up sample and the real structure used in application will be possible. Also how will the outline methodology be extended for use in real applications a small discussion on that will be useful. Especially in terms of the sensor placement on the real structure as well as processing of the data.

4) The clarity will be improved if a flowchart for the RAPID algorithm is provided.

5) In the numerical validation what was the effect of the reduction of the thickness from the opposite side?

6) For Figure 18, in addition to the qualitative comparison the quantitative compariosn willin the reduction of amplitude will be helpful

7) Why the square sensor placement was chosen. How were the number and their location/spacing determined.

8) please ensure that all the units used are in the SI system or are consistent.

9) The sampling frequency for numerical analysis appears insufficient. The Figure 6 shows insufficient sample points along the time co-ordinate

10) line 125 7 symbols are used and only 6 descriptions are provided.

11) No mention that the paper is an extension of the paper from APWSHM has been made. The paper from APWSHM has not even been cited. 

I will encourage the authors to restructure the manuscript and expand on the discussions.

Author Response

Dear, Reviewer. 

I appreciate your advice and comment. 

We tried to fix what you ask to correct and edit part.

Most of the parts of the manuscript are fixed and edited.

It is really helpful and considerable works for this research work.

Probably, some edited comments and supplements might not be satisfied with your opinion.

Regardless of this problem, please give us good pieces of advice and opinion for improvement.

Thank for reviewing the manuscript.

Sincerely yours

Author: Yonghee Lee, Younho Cho.

Round  2

Reviewer 1 Report

Many of my previous comment were totally ignored in the manuscript. I have to repeat them:

1) In the abstract it promises: "The proposed SHM technique relies on sensors and therefore, it can be placed on the structure permanently and can monitor either passively or actively." In fact, Rayleigh and Lamb waves have a distinct propagation direction. The transmitter and receiver have to point to each other. The transducers cannot be in a fixed position for all measurements. Example: for transmission 1 -> 6 the transducer 1 has a different orientation than for the transmission 1 -> 7, or 1 -> 8 etc. Hence the proposed measurement cannot be done with fixed transducers.

2) Transmitter and receiver have to point exactly to each other (wedges!). How is this adjustment experimentally achieved? How long took such a complete data acquisition for one tomogram

Furthermore:

p 4, line 127: add in the manuscript a clear justification why these specific values were chosen (transducer frequencies, specimen thickness)

p 4, first sentence after eq. 4: Give here values for specimen thickness in this work

p 5, Fig. 2: give units; give formula of power (add into the manuscript, not only in response to reviewer)

p 8, Fig. 8: Labels of color bar is not readable

p 11, lines 276/277: the Rayleigh wave length is 6 mm, but it should be smaller than the thickness (6 mm): this means that the prerequisite (smaller than thickness) is not fulfilled. In line 276 you write that the frequency of the Rayleigh wave is 500 kHz, but in Fig. 14 you write it is 1 MHz: inconsistency

Author Response

Dear, Reviewer. 

I appreciate your advice and comment. 

We tried to fix everything, what you ask to correct and edit part.

Most of the parts of the manuscript are fixed and edited.

It is really helpful and considerable works for this research work.

Probably, some edited comments and supplements might not be satisfied with your opinion.

Regardless of this problem, please give us good pieces of advice and opinion for improvement.

Thank for reviewing the manuscript.

Sincerely yours

Authors: Yonghee Lee ([email protected]) and Younho Cho ([email protected])

Reviewer 4 Report

The authors significantly improved the manuscript according to my remarks. However, since the wavelength tunning applied for selective mode excitation is quite important, an additional explanation should be included in the paper (not only as a part of a response to the reviewer). In my opinion after this modification, the paper can be accepted.

Author Response

Dear, Reviewer. 

I appreciate your advice and comment. 

We tried to fix what you ask to correct and edit parts, properly.

Most of the parts of the manuscript are fixed and edited.

It is really helpful and considerable works for this research work.

Probably, some edited comments and supplements might not be satisfied with your opinion.

Regardless of this problem, please give us good pieces of advice and opinion for improvement.

Thank for reviewing the manuscript.

with best regards,

Authors: Yonghee Lee([email protected]) and Younho Cho([email protected])

Reviewer 5 Report

Some of the comments have not been addressed and some of the responses are not satisfactory.

For in stance the point 7 has not been addressed satisfactorily and the point made by the reviewer and the response do not make sense.

Similarly the response to point 5 is not phrased correctly and hence is not understood by the reviewer.

In general the writing carried out needs extensive editing before the review is possible

line 89-

line 101 and many more

Also in line 101 elastic constant variation is kind of contradictory thought and needs to be rephrased correctly.

I expect the authors to spend some time on the manuscript and the comments by all reviewers and address all of them well before a re-submission. 

Author Response

(The authors gave the same response as above.)
